# Quality of first antenatal care visits and perinatal outcomes: Evidence from a cohort study in Ethiopia, Kenya, South Africa, and India

Wen-Chien Yang[1], Shalom Sabwa[2,3], Anagaw Derseh Mebratie[4], Beatrice Amboko[5], Irene Mugenya[5], Sein Kim[2], Emily R. Smith[1], Monica Chaudhry[6], Nokuzola Cynthia Mzolo[7], Nompumelelo Gloria Mfeka-Nkabinde[8], Theodros Getachew[2,9], Tefera Taddele[9], Damen Haile Mariam[4], Sailesh Mohan[6], Prashant Jarhyan[10], Margaret E. Kruk[2,11], Catherine Arsenault[1]*

1 Department of Global Health, Milken Institute School of Public Health, The George Washington University, Washington, District of Columbia, United States of America, 2 Department of Global Health and Population, Harvard T. H. Chan School of Public Health, Boston, Massachusetts, United States of America, 3 Department of Epidemiology, Rollins School of Public Health, Emory University, Atlanta, Georgia, United States of America, 4 Department of Health Systems Management and Health Policy, School of Public Health, College of Health Sciences, Addis Ababa University, Addis Ababa, Ethiopia, 5 Health Economics Research Unit, KEMRI-Wellcome Trust Research Programme, Nairobi, Kenya, 6 Public Health Foundation of India, Delhi, New Delhi, India, 7 School of Nursing and Public Health, University of KwaZulu-Natal, Durban, South Africa, 8 Department of Family Medicine, Howard College Campus, University of KwaZulu-Natal, KwaZulu-Natal, South Africa, 9 Health System and Reproductive Health Research Directorate, Ethiopian Public Health Institute, Addis Ababa, Ethiopia, 10 Centre for Chronic Disease Control, Delhi, New Delhi, India, 11 Division of General Medicine & Geriatrics, John T. Milliken Department of Medicine, School of Medicine, Washington University in St. Louis, St. Louis, Missouri, United States of America

* catherine.arsenault@gwu.edu

## Abstract

Antenatal care (ANC) is crucial for maternal and newborn health. Although ANC coverage has improved globally, ANC quality remains suboptimal. This study assessed the quality of first ANC visits and their association with perinatal outcomes. We used data from the eCohort study that collected longitudinal data on ANC utilization and quality until the end of pregnancy in Ethiopia, Kenya, South Africa, and India. Perinatal outcomes of interest included fetal losses (miscarriages ≥13 weeks of gestation or stillbirths) and low birth weight (LBW) newborns. Good quality ANC at first visits was defined as receiving six essential care components: blood pressure measurement, blood and urine tests, ultrasound, iron and folic acid supplementation, and counseling on pregnancy danger signs. We conducted mixed-effect logistic regressions to assess associations between good quality ANC and perinatal outcomes, with sensitivity analyses where good quality ANC excluded ultrasound scans. Among 3,600 pregnant women followed until the end of pregnancy, 5.8% received all six components at their first visits (from 1.3% in India to 14.0% in Ethiopia), and 30.7% received five components (5.7% in India to 52.5% in Kenya). Between 3.7% of women in Ethiopia and 6.3% in India experienced a fetal loss. Between 8.5% of newborns in Ethiopia and 16.3% in India were born LBW. Good quality ANC at first visits was

**Data availability statement:** The analytic datasets are available from: https://dataverse.harvard.edu/dataset.xhtml?persistentId=doi:10.7910/DVN/Q0YKOT.

**Funding:** Funding for the eCohort study was obtained from the Bill and Melinda Gates Foundation (grant number INV-005254 to MEK) and the Swiss Federal Department of Foreign Affairs (grant number 81067262 to MEK). The funders had no role in study design, data collection and analysis, decision to publish, or preparation of the manuscript.

**Competing interests:** The authors have declared that no competing interests exist.

associated with a 32% to 59% lower risk of fetal loss (primary analysis: risk ratio (RR) 0.41, 95% confidence interval (CI) 0.10–0.72; sensitivity analysis: RR 0.68, 95% CI 0.44–0.92). No significant associations were observed between good quality ANC and LBW. This study identified important gaps in ANC quality and found that good quality ANC is associated with a lower fetal loss risk. Investments must be made to improve ANC quality and ensure the delivery of essential care in pregnancy.

## Author summary

### Why was this study done?

- ANC utilization has increased substantially in LMICs; however, the quality of care received remains insufficient.

- Most prior research on the relationship between ANC and perinatal outcomes has focused on the number of visits rather than the quality of care provided.

- Although the global dialogue has gradually shifted toward ANC quality, limited evidence exists on the association between ANC quality and critical perinatal outcomes, particularly fetal loss and LBW.

### What did the researchers do and find?

- We used data from a longitudinal study in Ethiopia, Kenya, South Africa, and India that collected detailed information on ANC and perinatal outcomes.

- Quality of the first ANC visit was poor overall. Only 6% of pregnant women received all six essential care components (blood pressure measurement, blood and urine tests, ultrasound examination, iron and folic acid supplementation, and counseling on pregnancy danger signs), and only 31% received five components (excluding ultrasound).

- Good-quality ANC at the first visit was significantly associated with a lower risk of fetal loss (late miscarriage and stillbirth), while no significant association was found with LBW.

### What do these findings mean?

- Poor-quality ANC is not only inefficient but potentially harmful—an issue of growing concern as more women in LMICs gain access to ANC services.

- National efforts should prioritize the delivery of essential care components at ANC visits to ensure high-quality care and improve perinatal outcomes.

## Introduction

Antenatal care (ANC) is crucial for ensuring the health of pregnant women and their babies. ANC serves critical functions, including health promotion, screening and

diagnoses of diseases, management of maternal morbidities and pregnancy complications, and prevention of illnesses throughout pregnancy [1]. The first ANC visit is particularly critical, as it establishes the foundation for pregnancy care by enabling early risk identification, baseline health assessments, and timely initiation of preventive interventions [1]. Global and national ANC guidelines recommend a core set of services during the first ANC content: blood pressure measurement, blood and urine testing, screening for infections such as HIV and syphilis, iron and folic acid supplementation, counseling on pregnancy danger signs and birth preparedness, and risk stratification to determine whether the woman requires routine or specialized care. An early obstetric ultrasound before 24 weeks of gestation is also recommended by the WHO and by three of the four study countries to estimate gestational age, detect fetal anomalies, and identify multiple pregnancies [1]. The first visit thus serves as a unique entry point for screening and early detection of conditions such as hypertension, anemia, infections, and high-risk pregnancies that, if left unidentified, may lead to adverse perinatal outcomes. Despite these recommendations, the quality of care delivered at the first ANC visit remains poorly documented, and few studies have examined whether the completeness of care at this initial contact is associated with perinatal outcomes.

Previous research has linked ANC to reduced adverse neonatal outcomes. However, a large body of evidence has primarily focused on ANC utilization, or in other words, the number of ANC visits. Two meta-analyses concluded that at least one ANC visit reduced the risk of neonatal mortality by 34% to 39% in Asia and sub-Saharan Africa [2,3]. A study using 193 Demographic and Health Surveys (DHS) from 69 countries showed that at least one ANC visit was associated with reduced neonatal and infant mortality [4]. Studies have also demonstrated that ANC utilization has protective effects against fetal loss and low birth weight (LBW) [4–9]. Given the benefits of ANC, low- and middle-income countries (LMICs), where most maternal and newborn mortality and morbidities occur, have made tremendous efforts to increase ANC coverage. However, the quality of care remains insufficient, even for women who initiate ANC early in their pregnancy and meet the recommended number of visits [10–12].

Studies have examined the associations between ANC quality and neonatal mortality; however, most have focused on neonatal mortality, with relatively few addressing stillbirths and LBW newborns. A 2022 UNICEF report on stillbirths in LMICs indicated notable evidence gaps in estimating the overall effect of ANC quality on stillbirths [13]. Nearly two million babies are stillborn every year, 98% in LMICs, causing significant economic and psychosocial consequences [14–16]. Antepartum stillbirths are linked to several modifiable risk factors, including maternal infections, non-communicable diseases, poor nutrition, and lifestyle factors [15,17–20]. Hence, high quality ANC is crucial for addressing these risk factors. Despite the limited evidence, one study from Ghana found an association between good quality ANC and a reduced risk of stillbirth [8]; another study in Nairobi, Kenya, also demonstrated the protective effects of good quality ANC against stillbirth [7]. However, both studies have design limitations: the former used data from a cross-sectional population-based survey, which may be prone to recall bias and limitations in causal inference, while the latter retrospectively reviewed medical records with a small sample size.

Although the prevalence has declined slowly over the past two decades, LBW remains a significant global health concern, with an estimated prevalence of 14.7% in 2020, affecting approximately 19.8 million newborns worldwide [21]. LBW newborns are not only at a significantly higher risk of morbidity and mortality but also more likely to develop non-communicable diseases, such as hypertension and diabetes, in adulthood [22,23]. The recently introduced "small vulnerable newborns" framework integrates preterm infants, small-for-gestational-age newborns, and those with LBW, advocating for a comprehensive and integrative approach to addressing the burden of small newborns [24,25]. ANC should incorporate evidence-based interventions and essential care components to prevent and manage LBW. Some research suggested that ANC attendance and good quality ANC reduced LBW [4–6,9].

Critical evidence gaps remain in understanding the association between ANC quality and perinatal outcomes in LMICs. Hence, this study aimed to describe ANC quality, identify key care components of good quality ANC, and estimate its association with fetal loss and the risk of LBW in four LMICs: Ethiopia, Kenya, South Africa, and India.

## Methods

### Data source

We used data from the Maternal and Newborn Health (MNH) eCohort study, a longitudinal mixed-mode survey (face-to-face and phone surveys) on maternal and newborn health care quality in four countries (Ethiopia, Kenya, South Africa, and India). The study was conducted at two sites in each country. Between 21 and 29 facilities were selected for inclusion in each country. The health facilities selected included 1) both public and private primary and secondary facilities in Ethiopia and Kenya, 2) only public primary and secondary facilities in India, and 3) only public primary facilities in South Africa [26]. Pregnant women aged 15 and above (or 18 and above in India), at any gestational age, who presented to these facilities to have their first ANC visit, planning to give birth in the study site and consenting to participate were enrolled in this study. The MNH eCohort study aimed to enroll 500 pregnant women per site in each county, with a minimum of 50 women per facility strata. Details of the MNH eCohort study design and methodology, including a description of study sites and recruitment, are available elsewhere [26–28]. The enrollment surveys conducted between April 2023 and January 2024 included in-person health assessments and a review of maternal health cards. The follow-up surveys were conducted via phone approximately every 4 weeks during pregnancy, and after delivery or the end of pregnancy was reported. Surveys covered the number of ANC visits, the content of care received, the participant's health, and the newborn's health if the pregnancy resulted in a live birth. The analytic sample for the present study included participants who were followed until the end of pregnancy and excluded women who were lost to follow-up as well as those who reported early miscarriages (pregnancy loss before 13 weeks of gestation) since miscarriages in the first trimester are generally caused by chromosomal abnormalities and are rarely preventable [29,30].

### Measures

We included two primary outcomes: fetal loss and LBW newborns. Fetal loss was defined as any pregnancy loss after 13 weeks of gestation, including late miscarriages and stillbirths. Late miscarriages were defined as pregnancy losses between 13 and 28 weeks of gestation, and stillbirths were defined as pregnancy losses after 28 weeks of gestation (excluding those born alive) in accordance with the WHO case definition [31]. Newborns' birth weights were self-reported by mothers. For women who did not know the actual birth weight, we used a survey question from the Demographic and Health Survey that asked women to recall the baby's birth size, including whether the baby was "very large, larger than average, average, smaller than average, or very small" at birth. Our primary case definition of LBW included newborns with birth weights less than 2.5 kg and newborns described as "smaller than average" or "very small" if actual birth weight data were unavailable [32]. Data on birth body weight or mother-reported birth body size were only collected for newborns who were still alive at the time of the final follow-up phone survey. Gestational age (GA) at baseline was used to exclude early miscarriages (<13 weeks) from the analyses and to distinguish stillbirths and miscarriages. The GA was estimated from the maternal health card where possible, using the recorded gestational age, date of last menstrual period (LMP), or estimated due date from the first ANC visit. When these were unavailable, LMP as reported by the woman was used, or, as a last resort, her self-reported weeks of pregnancy at baseline. GA was further categorized into trimesters, with the first trimester defined as <13 weeks of gestation, the second trimester defined as ≥13 to <28 weeks, and the third trimester defined as ≥28 weeks.

### Quality of first antenatal care visits

The primary independent variable was good quality ANC at the first visit, a binary variable indicating the receipt of six essential care components at the first ANC visit: blood pressure (BP) was measured, blood test performed (either a blood draw or finger prick), urine test performed, ultrasound examination, iron and folic acid (IFA) given or prescribed, and counseling on pregnancy danger signs. These six essential care components were chosen based on WHO recommendations

and previous studies on ANC quality [1,5,7,8,33,34]. We also created a good quality ANC binary variable based on the receipt of five care components (excluding ultrasound scan). We chose to exclude ultrasound scans because the South African ANC guidelines at the time of the study did not recommend them for routine care. In addition, the WHO recommends at least one ultrasound scan before 24 weeks of gestation. Therefore, women initiating ANC early might receive the ultrasound in subsequent visits before 24 weeks of gestation. Although the eCohort study obtained information on the content of follow-up visits, we restricted the care components to those received at the first visit to avoid 'immortal time bias' since pregnant women who experience a miscarriage or stillbirth after the first visit cannot obtain additional ANC. Including data from subsequent visits would bias the association upward and overestimate the protective effect of ANC [35].

## Covariates

We included independent variables potentially associated with the primary outcomes, including maternal age, marital status, whether the pregnancy was intended, education level (categorized into three groups: no education or some primary education, complete primary education, and complete secondary education or higher), health literacy (defined as answering six health knowledge questions correctly), and household wealth (based on ownership of certain assets and categorized into country-specific wealth tertiles) [26]. Second, we included a series of covariates related to the pregnant woman's health at baseline: self-rated health (very good or excellent, compared to good, fair, and poor), reporting any danger signs in pregnancy at baseline (vaginal bleeding, fever, fainting or loss of consciousness), and having risk factors at baseline (categorized by the number of risk factors: no risk factor, one risk factor, two risk factors, and three or more risk factors). The risk factors assessed included any chronic illness(es) known before the pregnancy, a history of obstetric complications (including Cesarean sections, preterm birth, stillbirth, neonatal death, or postpartum hemorrhage), HIV, and multiple pregnancy detected at the first ANC visit.

## Statistical analysis

First, we present descriptive characteristics of pregnant women at baseline and the proportion of women who received all six and all five and each of the six care components at the first visit in each country. Second, we reported the prevalence of fetal and neonatal outcomes (late miscarriage, stillbirth, and LBW). Third, we conducted mixed-effect logistic regressions to investigate the association between good quality ANC and fetal loss and LBW, respectively. Regressions included country-fixed effects, two-level random intercepts (for the study site and the health facility where the woman attended ANC), and robust standard errors. We used predictions from the fitted models to calculate marginal probabilities and obtained risk ratios for the association between good quality ANC at the first visit and each primary outcome.

Regressions were performed at the fetal and neonatal level, with all eligible fetuses/babies from multiple pregnancies included separately. The first model assessed associations between good quality ANC and fetal losses (late miscarriages ≥13 weeks of gestation or stillbirths). The fetal loss regression was restricted to women who were enrolled in their first trimester of pregnancy to avoid selection bias. Including pregnant women who were already in the second or third trimester at baseline would bias findings since pregnancies that have progressed to the second or third trimester are less likely to develop late miscarriages (which by definition can only occur in the second trimester between 13 and 28 weeks). In other words, participants who would have passed the period during which particular outcomes of interest could occur should be excluded from the sample. The second regression model assessed associations between good quality ANC and LBW. The LBW regression model included all women (regardless of pregnancy stage at enrollment) but controlled for GA (in trimester) at the first ANC visit.

We performed three sensitivity analyses. First, for fetal losses, we restricted the sample to women with more reliable baseline GA, defined as a baseline GA calculated based on LMP or EDD, and excluded women whose baseline GA was self-reported based on the number of weeks they thought to be pregnant. Second, we conducted a sensitivity analysis for

 

LBW newborns that restricted the sample to those newborns with actual birth weight data, excluding those with birth body sizes described as "smaller than average or very small." In addition, each regression model (including sensitivity analyses) was repeated using the independent variable of good quality ANC with five components that excluded ultrasounds. Third, we run models where ANC quality was measured by a continuous score ranging from 0 to 100%, reflecting the proportion of essential items out of the six received by pregnant women. Detailed descriptions of models were shown in **Table A** in S1 Appendix. Analyses were conducted using complete data, excluding cases with missing values. Statistical significance was determined based on a p-value < 0.05, and 95% confidence intervals were provided. All analyses were performed using STATA version 18.0.

### Ethical approval

The study protocol was reviewed and approved by the Institutional Review Boards (IRB) of the Harvard T.H. Chan School of Public Health (protocol #IRB22–0487), the Kenya Medical Research Institute (protocol number KEMRI/SERU/CGMR-C/4226), the Ethiopian Public Health Institute (protocol number EPHI-IRB-448–2022), the University of KwaZulu-Natal (protocol number BREC/00004645/2022), and the Public Health Foundation of India (protocol number TRC-IEC 495/22). Participants were recruited for the study from 03/April/2023-19/May/2023 in Ethiopia, from 9/Oct/2023-26/Jan/2024 in India, from 12/Jun/2023-01/Sep/2023 in Kenya, and from 24/Apr/2023–6/Sep/2023 in South Africa. Formal informed consent was obtained from all adult study participants and emancipated minors, or from formal guardians or next-of-kin for study participants under the age of 18 years old in accordance with local regulations.

## Results

### Characteristics of study participants

This study included 3,600 women followed from their first ANC visit until the end of pregnancy (**Table 1**). The mean age of participants was 25.9 years old (mean age 24.5 in India to 26.9 in Kenya), with more adolescents enrolled in South Africa. Nearly all participants in Ethiopia (97.7%) and India (99.9%) and the majority in Kenya (78.5%) were married or partnered, compared to a small proportion in South Africa (11.7%). Pregnancy intention varied widely, with 91.0% in India, 73.4% in Ethiopia, 60.9% in Kenya, and only 16.9% in South Africa reporting the current pregnancy as intended. Maternal health status at baseline varied. Around 60% of women in India (66.6%) and Kenya (56.0%) rated their health as very good or excellent, compared to about 40% in South Africa and Ethiopia. In India, 19.1% of women had at least one risk factor, followed by 24.8% in Ethiopia and 28.2% in Kenya; in contrast, this proportion was markedly higher in South Africa at 50.2%, where HIV prevalence was the highest (28.8%). Few women in India (5.5%) and Kenya (8.3%) reported any pregnancy danger signs at baseline, compared to 13.1% in Ethiopia and 25.6% in South Africa. Half of the women in India initiated ANC visits in the first trimester, compared to only 13.8% to 38.0% in the three sub-Saharan African countries. In addition, there were discrepancies between multiple pregnancies detected at the first ANC visit and identified at delivery.

### Antenatal care quality at the first visit

Only 5.8% of women received all six care components at their first ANC visit, ranging from 1.3% only in India to 14.0% in Ethiopia (**Table 2**). Use of ultrasound was low, ranging from only 7.6% of women in South Africa to 43.4% in Ethiopia. Counseling on pregnancy danger signs was highest in Kenya (60.9%), followed by 49.4% in South Africa, 31.0% in Ethiopia, and lowest in India (14.9%). Iron and folic acid supplements were given or prescribed to most women, from 80.8% in Ethiopia to 93.1% in Kenya. Excluding ultrasounds increased the proportion of women who received all five care components to 30.7%. Results stratified by research sites and by facility ownership in each country are shown in **Table B** in S1 Appendix and **Table C** in S1 Appendix. When excluding private facilities, the proportion who received all six care components ranged from 1.3% in India to 6.9% in Kenya.

Table 1. Characteristics of pregnant women followed until the end of pregnancy included in the study by country[1].

| | Overall N = 3600 | Ethiopia N = 883 | Kenya N = 896 | South Africa N = 854 | India N = 967 |
|---|---|---|---|---|---|
| | Frequency (%) | Frequency (%) | Frequency (%) | Frequency (%) | Frequency (%) |
| **Sociodemographic** | | | | | |
| Age, years (mean/standard deviation) | 25.9 (0.1) | 25.6 (0.2) | 26.9 (0.2) | 26.7 (0.2) | 24.5 (0.1) |
| Age category, years | | | | | |
| 15 to <20 | 390 (10.8%) | 73 (8.3%) | 104 (11.6%) | 142 (16.6%) | 71 (7.3%) |
| ≥20 to <35 | 2887 (80.2%) | 753 (85.3%) | 667 (74.4%) | 591 (69.2%) | 876 (90.6%) |
| ≥35 | 323 (9.0%) | 57 (6.5%) | 125 (14.0%) | 121 (14.2%) | 20 (2.1%) |
| Married or partnered | 2628 (73.1%) | 862 (97.7%) | 700 (78.5%) | 100 (11.7%) | 966 (99.9%) |
| Intended pregnancy | 2214 (61.8%) | 648 (73.4%) | 546 (60.9%) | 144 (16.9%) | 876 (91.0%) |
| Education level | | | | | |
| No education or some primary | 557 (15.5%) | 334 (37.8%) | 65 (7.3%) | 6 (0.7%) | 152 (15.7%) |
| Complete primary | 1224 (34.0%) | 353 (40.0%) | 313 (34.9%) | 248 (29.1%) | 310 (32.1%) |
| Complete secondary or higher | 1817 (50.5%) | 196 (22.2%) | 518 (57.8%) | 599 (70.2%) | 504 (52.2%) |
| Health literacy[2] | 1032 (28.7%) | 237 (26.8%) | 398 (44.4%) | 277 (32.4%) | 120 (12.4%) |
| Wealth status[3] | | | | | |
| Poorest | 1129 (31.6%) | 283 (32.7%) | 299 (33.6%) | 228 (26.7%) | 319 (33.2%) |
| Middle | 1395 (39.1%) | 286 (33.1%) | 383 (43.0%) | 339 (39.7%) | 387 (40.3%) |
| Richest | 1045 (29.3%) | 296 (34.2%) | 209 (23.5%) | 286 (33.5%) | 254 (26.5%) |
| **Maternal health** | | | | | |
| Self-rated own health as very good or excellent[4] | 1843 (51.2%) | 335 (37.9%) | 502 (56.0%) | 362 (42.4%) | 644 (66.6%) |
| Risk factor at baseline[5] | | | | | |
| No risk factor | 2513 (69.8%) | 664 (75.2%) | 643 (71.8%) | 425 (49.8%) | 782 (80.9%) |
| One risk factor | 797 (22.1%) | 184 (20.8%) | 192 (21.4%) | 280 (32.8%) | 142 (14.7%) |
| Two risk factors | 199 (5.5%) | 29 (3.3%) | 33 (3.7%) | 104 (12.2%) | 33 (3.4%) |
| Three or more risk factors | 91 (2.5%) | 6 (0.7%) | 28 (3.1%) | 45 (5.3%) | 10 (1.0%) |
| HIV status[6] | 277 (7.7%) | 13 (1.5%) | 15 (1.7%) | 244 (28.8%) | 5 (0.5%) |
| **Pregnancy characteristics** | | | | | |
| Trimester of the first ANC visit | | | | | |
| First trimester[7] | 1188 (33.3%) | 259 (29.9%) | 124 (13.8%) | 319 (38.0%) | 486 (50.3%) |
| Second trimester[7] | 2047 (57.4%) | 559 (64.6%) | 613 (68.4%) | 444 (52.9%) | 431 (44.6%) |
| Third trimester[7] | 334 (9.4%) | 48 (5.5%) | 159 (17.8%) | 77 (9.2%) | 50 (5.2%) |
| Report at least one danger sign at baseline[8] | 461 (12.8%) | 116 (13.1%) | 74 (8.3%) | 218 (25.6%) | 53 (5.5%) |
| Multiple pregnancy known at baseline | 24 (0.7%) | 7 (0.8%) | 4 (0.5%) | 5 (0.6%) | 8 (0.8%) |
| Multiple pregnancy identified at delivery | 59 (1.6%) | 16 (1.8%) | 18 (2.0%) | 19 (2.2%) | 6 (0.6%) |

[1]Participants who had an early miscarriage (fetal loss before 13 weeks of gestation) or were lost to follow-up after the first ANC visit were excluded.

[2]Defined as answering six health knowledge questions correctly. The six questions were adopted from the Indian Health and Human Development Survey.

[3]Wealth tertile is country specific.

[4]Self-rated own health was based on a Likert scale of five levels (excellent, very good, good, fair, and poor).

[5]Includes any chronic systemic illness(es) known before pregnancy (including diabetes, hypertension, cardiac disease, HIV, mental health disorder, schizophrenia, epilepsy, seizure, renal disorder, asthma, tuberculosis, anemia, hemoglobinopathy, chronic pelvic inflammatory disease, ovarian cyst, fibroids, uterine myoma, genital tract abnormalities, thyroid cancer, thyroid disease, peptic ulcer disease, gestational hypertension in previous pregnancy, a history of stroke) and any history of obstetric complications (Cesarean section, stillbirth, preterm birth, neonatal death, and postpartum hemorrhage).

[6]HIV was reported separately in this table while it was included in the development of categorization of risk factors.

[7]The first trimester is defined as less than 13 weeks of gestation; the second trimester is defined as between 13 to less than 28 weeks of gestation; the third trimester is defined as above 28 weeks of gestation.

[8]Includes vaginal bleeding, fever, and fainting or loss of consciousness.

**Table 2. Proportion of pregnant women who received each of the six and all six care components at first ANC visits by country.**

| ANC quality care components | Overall | Ethiopia | Kenya | South Africa | India |
|---|---|---|---|---|---|
| | N = 3600 | N = 883 | N = 896 | N = 854 | N = 967 |
| | Frequency (%) | Frequency (%) | Frequency (%) | Frequency (%) | Frequency (%) |
| **Physical examination** | | | | | |
| Blood pressure measurement | 3307 (91.9%) | 640 (72.5%) | 861 (96.2%) | 852 (99.8%) | 954 (98.8%) |
| **Test and examinations** | | | | | |
| Blood test (blood draw or finger prick) | 3422 (95.1%) | 838 (94.9%) | 876 (97.9%) | 853 (99.9%) | 855 (88.5%) |
| Urine test | 3048 (84.7%) | 752 (85.2%) | 801 (89.5%) | 850 (99.5%) | 645 (66.7%) |
| Ultrasound | 709 (19.8%) | 383 (43.4%) | 74 (8.3%) | 65 (7.6%) | 187 (19.5%) |
| **Counseling** | | | | | |
| Signs of pregnancy complications | 1380 (38.5%) | 273 (31.0%) | 544 (60.9%) | 420 (49.4%) | 143 (14.9%) |
| **Preventions** | | | | | |
| Iron and folic acid given or prescribed | 3220 (89.7%) | 713 (80.8%) | 831 (93.1%) | 790 (92.7%) | 886 (92.2%) |
| **Completeness of all six care components**[1] | 209 (5.8%) | 124 (14.0%) | 50 (5.6%) | 22 (2.6%) | 13 (1.3%) |
| **Completeness of all five care components**[2] | 1105 (30.7%) | 179 (20.3%) | 470 (52.5%) | 401 (47.0%) | 55 (5.7%) |

[1]Include blood pressure measurement, blood test, urine test, ultrasound examination, iron and folic acid given or prescribed, counseling on signs of pregnancy complications.

[2]Include all the care components except for ultrasound examination.

## Perinatal outcomes

Table 3 describes fetal and neonatal outcomes. The proportion of late miscarriages ranged from 2.3% in Ethiopia to 3.9% in India, while stillbirths ranged from 1.3% to 2.4%. Among neonates who were alive at the post-delivery follow-up survey, India had the highest rates of LBW infants (16.3%), compared to 8.5% to 13.1% in the other three countries. Data on EDD, LMP, and actual newborn birth weight was available for the majority of the observations in India, Kenya, and South Africa. However, in Ethiopia, data on EDD or LMP were available for only half (51.1%) of neonates, and actual newborn birth weight data were available for 59.6% of newborns.

## Results of mixed-effect logistic regressions

Risk ratios (RR) for the association between good quality ANC and perinatal outcomes from the eight regression models are summarized in Fig 1. Receiving all six ANC components was associated with a 58% reduced risk of fetal loss (late miscarriages or stillbirths, RR 0.41, 95% CI 0.10–0.72). Models with good quality ANC that excluded ultrasounds showed that the receipt of five essential care components was associated with reduced risks of fetal loss (RR 0.68, 95% CI 0.44–0.92 in primary analysis and RR 0.77, 95% CI 0.59–0.94 in sensitivity analysis). Regressions for LBW newborns, both primary and sensitivity analyses, did not show statistically significant associations between receiving good quality ANC and LBW newborns.

In addition, we found that pregnant women who reported any pregnancy danger signs at baseline had increased risks of fetal losses in both regression models where good quality ANC had six components (Odds ratio (OR) 1.87, 95% CI 1.25–2.80) and five components (OR 1.93, 95% CI 1.29–2.87) at the first visit (Table 4). In the regressions for LBW newborns, mothers giving birth in South Africa and India were more likely to have a LBW newborn compared to those in Ethiopia. Women aged 20–35 years old, compared to those younger than 20 years old, had significantly lower odds of delivering LBW newborns in both models where good quality ANC had six or five components at the first visit (OR 0.80, 95% CI 0.69–0.93; OR 0.80, 95% CI 0.69–0.94) (Table 4). Regressions using ANC quality as a continuous score did not show associations between ANC quality score and both outcomes of interest (Table D in S1 Appendix).

**Table 3. Fetal and neonatal outcomes.**

| | Overall | Ethiopia | Kenya | South Africa | India |
|---|---|---|---|---|---|
| | N = 3660 | N = 900 | N = 914 | N = 873 | N = 973 |
| **Fetal outcomes** | Frequency (%) | Frequency (%) | Frequency (%) | Frequency (%) | Frequency (%) |
| Late miscarriage[1] | 104 (2.8%) | 21 (2.3%) | 21 (2.3%) | 24 (2.8%) | 38 (3.9%) |
| Stillbirth[2] | 60 (1.6%) | 12 (1.3%) | 14 (1.5%) | 11 (1.3%) | 23 (2.4%) |
| Livebirth | 3496 (95.5%) | 867 (96.3%) | 879 (96.2%) | 838 (96.0%) | 912 (94.0%) |
| **Fetal outcomes (among those with more reliable baseline gestational age)[3]** | **N = 2925** (79.9%)[6] | **N = 460** (51.1%)[6] | **N = 875** (95.7%)[6] | **N = 868** (99.4%)[6] | **N = 722** (74.2%)[6] |
| Late miscarriage[1] | 80 (2.7%) | 9 (2.0%) | 20 (2.3%) | 24 (2.8%) | 27 (3.7%) |
| Stillbirth[2] | 48 (1.6%) | 6 (1.3%) | 13 (1.5%) | 11 (1.3%) | 18 (2.5%) |
| **Neonatal outcomes (among neonates still alive at the post-delivery survey)** | **N = 3441**[7] | **N = 851**[7] | **N = 862**[7] | **N = 828**[7] | **N = 900**[7] |
| Low birth weight newborn[4] | 398 (11.7%) | 72 (8.6%) | 73 (8.5%) | 106 (13.1%) | 147 (16.3%) |
| **Neonatal outcomes (among neonates still alive at the post-delivery survey and who had data on actual birth weight)** | **N = 3021** (87.8%)[8] | **N = 507** (59.6%)[8] | **N = 842** (97.7%)[8] | **N = 773** (93.4%)[8] | **N = 899** (99.9%)[8] |
| Low birth weight newborn[5] | 346 (11.5%) | 37 (7.3%) | 69 (8.2%) | 93 (12.0%) | 147 (16.4%) |

[1]Defined as fetal loss between 13 and 28 weeks of gestation.

[2]Defined as fetal loss after 28 weeks of gestation.

[3]More reliable baseline gestational age was determined by last menstrual period or estimated due date.

[4]Defined as newborn birth body weight less than 2.5 kg, or self-reported newborn size as smaller than average or very small if actual birth weight was unavailable.

[5]Defined as newborn birth body weight less than 2.5 kg.

[6]The percentages represent the proportions of fetuses born to mothers with more reliable gestational age dating, calculated as the number with a reliable baseline gestational age (determined by last menstrual period) divided by the total number of fetuses.

[7]Among neonates who were still alive at the post delivery survey, 13 (out of 851) in Ethiopia, 4 (out of 862) in Kenya, and 18 (out of 828) in South Africa had missing data for both birth weight and size.

[8]The percentages represent the proportion of neonates who were alive at post-delivery surveys and had birthweight data, calculated as the number with birthweight data divided by the number of neonates alive at post-delivery surveys, regardless of body weight or size information source (actual birth-weight or mothers' self-reported birth body size if weight data was unavailable).

## Discussion

In this study, we used data from the MNH eCohort study in four countries to investigate associations between the quality of first ANC visits and perinatal outcomes. We found that less than 10% of pregnant women in the study received all six key care components during their first ANC visit. While the majority of women had their blood pressure measured and gave blood and urine samples, the proportion of women receiving ultrasound examinations and being counseled on pregnancy danger signs was much lower. Our analysis also showed that good quality ANC, defined as receiving at least five or six care components during the first ANC visit, was associated with a reduced risk of fetal loss (≥13 weeks gestation) but not LBW. There are a few potential explanations for this lack of association with LBW. First, the care components we identified were proxies of good quality ANC drawn from the first ANC visit only and may not capture all essential elements that affect LBW. Second other evidence-based antenatal nutritional interventions proven to prevent LBW were not included in the study including balanced energy protein (BEP) for undernourished women and multiple micronutrient supplementation (MMS) [36,37]. MMS is not yet routinely provided in the study countries, though it is likely to be scaled up in the coming years. Similarly, BEP supplementation is not universally available. Programs such as Ethiopia's targeted supplementary

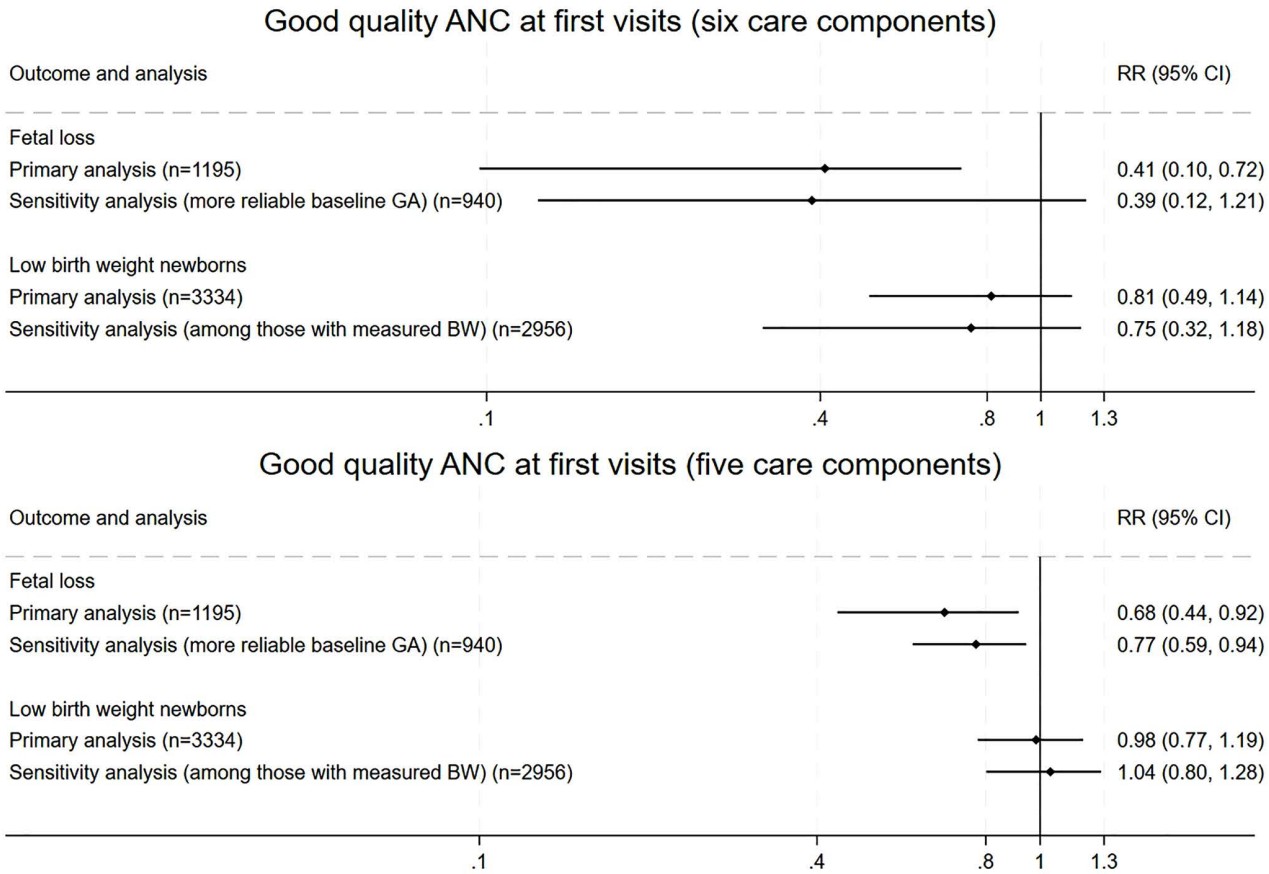

**Fig 1. Risk ratios (RR) of fetal loss (late miscarriage and stillbirth) and LBW newborns between those who received good quality ANC at the first visit and those who did not.** Results are from mixed-effect logistic regression models that assessed the associations of good quality ANC (defined as receiving six basic care components at first ANC visits or five, excluding ultrasound) and fetal loss (late miscarriage and stillbirth) and LBW newborns, respectively. The samples for fetal loss regressions came from 1) the fetuses with an outcome of either miscarriage, stillbirth, or livebirth, restricted to those whose mothers enrolled in the first trimester for primary analysis, and 2) the fetuses with an outcome of either late miscarriage, stillbirth, or live-birth, restricted to those whose mothers enrolled in the first trimester and had more reliable gestational age dating for secondary analysis. The samples for LBW regressions came from 1) the neonates who were alive at post-delivery surveys for primary analysis, and 2) the neonates who were alive at post-delivery surveys and had actual birthweight data for secondary analysis. All regressions included country as a fixed effect, study site and facility as random intercepts, and a list of covariates, including age, education level, health literacy, health status, pregnancy intention, risk factor categories, and presence of any pregnancy danger signs at first ANC visits. For LBW regression, time of enrolment at first ANC visits (in trimester) was also included as a covariate. GA: gestational age; LMP: last menstrual period; EDD: estimated due date; BW: birth weight; LBW: low birth weight.

feeding program deliver BEP to undernourished pregnant women (30%–50% of pregnant women are undernourished in Ethiopia), but implementation has faced considerable challenges, and only certain districts are targeted. IFA supplementation may also reduce LBW; however, receipt at the first visit does not necessarily reflect adherence to supplementation for at least 90 days. Improved ANC quality and better integration of nutritional interventions likely has the potential to further reduce LBW.

The association between good quality ANC and fetal losses confirms the importance of quality ANC. Notably, we observed larger effect sizes in models where good quality ANC was defined as receiving six care components compared to models that excluded ultrasounds. This specific finding may reflect the value of ultrasound scans. Obstetric ultrasounds, when performed early, allows for accurate gestational age assessment. Repeated scans enable monitoring of fetal

**Table 4. Results of mixed effect logistic regressions for fetal losses (late miscarriage and stillbirth) and low birth weight newborns.**

| | Fetal loss models | | | | | | Low birth weight newborn models | | | | | |
|---|---|---|---|---|---|---|---|---|---|---|---|---|
| | Good quality ANC (received six care components at the first visit) N = 1195 | | | Good quality ANC (received five care components at the first visit) N = 1195 | | | Good quality ANC (received six care components at the first visit) N = 3334 | | | Good quality ANC (received five care components at the first visit) N = 3334 | | |
| | Odds Ratio (OR) | 95% Confidence Interval (CI) | p-value | Odds Ratio (OR) | 95% Confidence Interval (CI) | p-value | Odds Ratio (OR) | 95% Confidence Interval (CI) | p-value | Odds Ratio (OR) | 95% Confidence Interval (CI) | p-value |
| **Received good quality ANC** | **0.38** | **0.17 – 0.85** | **0.019** | **0.65** | **0.43 – 0.97** | **0.035** | 0.79 | 0.50 – 1.24 | 0.305 | 0.98 | 0.77 – 1.26 | 0.877 |
| **Socioeconomic and demographic status** | | | | | | | | | | | | |
| *Country* | | | | | | | | | | | | |
| Ethiopia | *ref* | | | *ref* | | | *ref* | | | *ref* | | |
| Kenya | 0.81 | 0.46 – 1.42 | 0.457 | 0.98 | 0.49 – 1.99 | 0.961 | 0.97 | 0.55 – 1.73 | 0.926 | 0.99 | 0.55 – 1.78 | 0.986 |
| South Africa | 0.50 | 0.16 – 1.54 | 0.228 | 0.61 | 0.18 – 2.08 | 0.433 | **1.68** | **1.23 – 2.29** | **0.001** | **1.73** | **1.21 – 2.47** | **0.002** |
| India | 1.19 | 0.67 – 2.12 | 0.555 | 1.23 | 0.66 – 2.27 | 0.516 | **1.96** | **1.28 – 2.99** | **0.002** | **2.01** | **1.24 – 3.24** | **0.004** |
| *Age category. years* | | | | | | | | | | | | |
| 15 to <20 | *ref* | | | *ref* | | | *ref* | | | *ref* | | |
| ≥20 to <35 | 0.67 | 0.42 – 1.07 | 0.097 | 0.67 | 0.42 – 1.08 | 0.101 | **0.80** | **0.69 – 0.93** | **0.004** | **0.80** | **0.69 – 0.94** | **0.005** |
| ≥35 | 1.39 | 0.39 – 4.91 | 0.607 | 1.38 | 0.39 – 4.94 | 0.616 | 1.06 | 0.70 – 1.60 | 0.777 | 1.06 | 0.70 – 1.59 | 0.786 |
| *Intended pregnancy* | 0.91 | 0.48 – 1.71 | 0.762 | 0.91 | 0.47 – 1.73 | 0.766 | 1.15 | 0.80 – 1.66 | 0.441 | 1.16 | 0.80 – 1.67 | 0.430 |
| *Education level* | | | | | | | | | | | | |
| No education or some primary | *ref* | | | *ref* | | | *ref* | | | *ref* | | |
| Complete primary | 1.18 | 0.68 – 2.03 | 0.560 | 1.15 | 0.65 – 2.04 | 0.623 | **1.40** | **1.05 – 1.87** | **0.021** | **1.40** | **1.05 – 1.87** | **0.024** |
| Complete secondary or higher | 1.07 | 0.54 – 2.11 | 0.854 | 1.05 | 0.53 – 2.10 | 0.885 | 1.02 | 0.77 – 1.36 | 0.891 | 1.02 | 0.76 – 1.36 | 0.895 |
| *Health literacy[1]* | 1.59 | 0.92 – 2.77 | 0.098 | 1.61 | 0.93 – 2.78 | 0.089 | 0.87 | 0.74 – 1.02 | 0.091 | 0.87 | 0.74 – 1.02 | 0.090 |
| *Wealth* | | | | | | | | | | | | |
| Poorest | *ref* | | | *ref* | | | *ref* | | | *ref* | | |
| Middle | 0.85 | 0.55 – 1.31 | 0.455 | 0.86 | 0.56 – 1.31 | 0.481 | 0.97 | 0.67 – 1.40 | 0.868 | 0.97 | 0.67 – 1.41 | 0.867 |
| Richest | 0.83 | 0.64 – 1.07 | 0.158 | 0.83 | 0.66 – 1.06 | 0.139 | 0.82 | 0.48 – 1.42 | 0.478 | 0.82 | 0.47 – 1.42 | 0.476 |
| **Trimester of the first ANC visit** | | | | | | | | | | | | |
| First trimester[2] | | | | | | | *ref* | | | *ref* | | |
| Second trimester[2] | | | | | | | 0.96 | 0.80 – 1.16 | 0.705 | 0.97 | 0.81 – 1.16 | 0.711 |
| Third trimester[2] | | | | | | | 0.81 | 0.41 – 1.60 | 0.549 | 0.82 | 0.42 – 1.60 | 0.552 |
| **Maternal health** | | | | | | | | | | | | |
| *Self-rated own health as very good or excellent[3]* | 1.06 | 0.58 – 1.96 | 0.839 | 1.07 | 0.59 – 1.94 | 0.818 | 0.85 | 0.67 – 1.08 | 0.195 | 0.85 | 0.67 – 1.08 | 0.194 |
| *Risk factor at baseline[4]* | | | | | | | | | | | | |
| No risk factor | *ref* | | | *ref* | | | *ref* | | | *ref* | | |
| One risk factor | 1.41 | 0.87 – 2.30 | 0.167 | 1.44 | 0.88 – 2.34 | 0.145 | 1.17 | 0.96 – 1.44 | 0.125 | 1.18 | 0.96 – 1.44 | 0.121 |
| Two risk factors | 1.51 | 0.63 – 3.61 | 0.356 | 1.47 | 0.64 – 3.35 | 0.362 | 1.19 | 0.77 – 1.84 | 0.428 | 1.18 | 0.77 – 1.82 | 0.441 |
| Three or more risk factors | 2.48 | 0.94 – 6.55 | 0.066 | **2.68** | **1.05 – 6.82** | **0.039** | 0.89 | 0.38 – 2.08 | 0.794 | 0.90 | 0.39 – 2.09 | 0.807 |

*(Continued)*

**Table 4.** (Continued)

| | Fetal loss models | | | | | | Low birth weight newborn models | | | | | |
|---|---|---|---|---|---|---|---|---|---|---|---|---|
| | Good quality ANC (received six care components at the first visit) N = 1195 | | | Good quality ANC (received five care components at the first visit) N = 1195 | | | Good quality ANC (received six care components at the first visit) N = 3334 | | | Good quality ANC (received five care components at the first visit) N = 3334 | | |
| | Odds Ratio (OR) | 95% Confidence Interval (CI) | p-value | Odds Ratio (OR) | 95% Confidence Interval (CI) | p-value | Odds Ratio (OR) | 95% Confidence Interval (CI) | p-value | Odds Ratio (OR) | 95% Confidence Interval (CI) | p-value |
| *Reported any pregnancy danger sign at baseline*[5] | 1.87 | 1.25 – 2.80 | 0.002 | 1.93 | 1.29 – 2.87 | 0.001 | 0.83 | 0.56 – 1.21 | 0.329 | 0.82 | 0.56 – 1.20 | 0.307 |

[1]Defined as answering six health knowledge questions correctly. The six questions were adopted from the Indian Health and Human Development Survey.

[2]The first trimester is defined as less than 13 weeks of gestation; the second trimester is defined as between 13 and less than 28 weeks of gestation; the third trimester is defined as above 28 weeks of gestation.

[3]Self-rated own health was based on a Likert scale of five levels (excellent, very good, good, fair, and poor)

[4]Risk factors at baseline include any chronic systemic illness(es) known before pregnancy (including diabetes, hypertension, cardiac disease, HIV, mental health disorder, schizophrenia, epilepsy, seizure, renal disorder, asthma, tuberculosis, anemia, hemoglobinopathy, chronic pelvic inflammatory disease, ovarian cyst, fibroids, uterine myoma, genital tract abnormalities, thyroid cancer, thyroid disease, peptic ulcer disease, gestational hypertension in previous pregnancy, a history of stroke), any history of obstetric complications (Cesarean section, stillbirth, preterm birth, neonatal death, and postpartum hemorrhage), and multiple pregnancy known at the first ANC visit

[5]Pregnancy danger signs at baseline include vaginal bleeding, fever, and fainting or loss of consciousness.

growth and detection of placental abnormalities linked to stillbirth, identification of congenital anomalies, and recognition of breech presentation—all of which are crucial for maternal and newborn health [38,39]. While we found low receipt of ultrasound scans at first visits, we acknowledged that all four countries recommend an ultrasound scan before 24 weeks of gestation, which might occur after the first visit. Since 90% of women in our study started ANC visits in the first and second trimesters, many may not receive a scan at the first visit based on national recommendations. Low ultrasound coverage may also explain why multiple pregnancies were rarely detected at the first visit in our study. In South Africa, for example, few women received an ultrasound during their first ANC visit, and the majority (74%) of multiple pregnancies went undetected at that stage. This is particularly concerning, as multiple pregnancy is a well-established risk factor for numerous adverse outcomes [40]. Early identification through obstetric ultrasound facilitates risk stratification and timely referral. However, critical gaps in ultrasound use persist in LMICs: access to equipment remains limited [41,42]; and scans are often primarily diagnostic, with the capacity to translate findings into effective management strategies varying considerably depending on health provider competence [43]. A 2014 cluster randomized trial assessed the effects of two ultrasound scans in pregnancy, one at 16–22 weeks and one at 32–36 weeks, on maternal and neonatal mortality in five LMICs (Zambia, Kenya, the Democratic Republic of the Congo, Pakistan, and Guatemala). The study showed an anticipated higher detection of multiple pregnancies but no improvement in maternal, fetal, and neonatal survival [43]. In our study, assessing early ultrasound scans in the first ANC visit, an aspect overlooked in prior research, offers critical insights [1]. Our work contributes to the growing evidence on the benefits of ANC quality in the context of increasing ANC coverage [10–12,33,44]. In a study using household surveys from 91 LMICs, only two-thirds of women who used ANC received three essential care components (BP measurement, blood test, and urine test) [12]. Many pregnant women in low-resource countries who receive the recommended number of ANC visits do not receive essential care content [10,11]. A study across 10 LMICs found that while most women in need of ANC had at least one visit, only two-fifths attended four or more. In addition, the quality of care was suboptimal, with blood pressure measurement being the most commonly performed service, while counseling on complications was the least provided [10]. A population-based study in India found

poor coverage of quality-adjusted ANC, defined as receiving BP and weight measurements, abdomen examinations, and blood and urine tests [33]. Our results reinforce existing evidence on the insufficient quality of ANC, urging global efforts to prioritize the delivery of necessary care and evidence-based interventions at ANC visits rather than merely increasing the number of ANC contacts. In addition to poor quality ANC, another interesting finding is the significant association between self-reported danger signs and fetal loss at baseline. Future research should explore the quality of ongoing management of danger signs that may lead to severe pregnancy complications.

In our study, between 8.5% to 16.3% of babies were LBW. However, the actual proportion of LBW in our sample may be higher, as birth weight or size data were unavailable those who died. In addition, there was a higher rate of LBW in South Africa, which might be partly attributable to a higher number of adolescent pregnancies and HIV infections. LBW babies, together with preterm and SGA newborns–collectively termed "small vulnerable newborns"–are at increased risk not only of mortality throughout the first year of life but also of suboptimal neurodevelopment [45,46].

The present study had limitations. First, our analyses and inference testing are limited by the low incidence of outcomes of interest. Late miscarriages and stillbirths are rare outcomes; only 4.4% of women in our sample experienced them, leading to large standard errors. Similarly, some sensitivity analyses were likely underpowered due to small sample sizes. Second, our results may have limited generalizability as the study enrolled participants from two particular sites in each country, not nationally representative samples. Third, we only included care components at the first ANC visit to define good quality. These components, while fundamental, are surrogate indicators for the overall ANC quality throughout pregnancy. The rationale for not including care content at follow-up visits was to avoid immortal time bias. This bias arises when the exposure definition encompasses a period during which the outcome cannot occur. In our study, if we had defined good-quality ANC based on care received across multiple visits, women who experienced a miscarriage or stillbirth early in pregnancy would have had fewer visits—and thus fewer opportunities to receive care—simply because their pregnancies ended sooner. Conversely, women with ongoing pregnancies would have appeared to receive higher-quality care, not necessarily because their care was better, but because they survived long enough to attend additional visits. This would create a spurious protective association between ANC quality and fetal survival, overestimating the effect of care received. By restricting the exposure to care components at the first visit only, we ensured that the assessment of ANC quality was not contingent on pregnancy duration. In perinatal epidemiology studies, this bias can significantly affect analyses examining the effects of pregnancy exposures on stillbirths or miscarriages [35,47]. Fourth, newborn birth weight and size data were only collected for babies alive at the post-delivery follow-up survey, excluding the birth weight data for neonatal or infant deaths. As LBW babies are at a higher risk of death compared to normal-weight babies, some cases of neonatal and infant deaths likely had low birth weights but were excluded from the analysis. Fifth, the LBW analysis was limited by the mother's self-reported baby birth size, with only 60% of babies in Ethiopia having actual birth weight data, compared to more than 90% in the other three countries. In addition, we were unable to distinguish between antepartum and intrapartum stillbirths in the fetal loss analysis. Lastly, these countries differed in their national ANC guidelines. For instance, Ethiopia and Kenya recommend an ultrasound scan for all women before 24 weeks of gestation, while South Africa's 4th edition national ANC guidelines recommend an ultrasound scan only for women who are unsure about their LMP (and the 5th edition guidelines have changed to recommend a scan for everyone). This might explain the low rate of receiving ultrasound scans in South Africa. Lastly, we acknowledge the limited generalizability of our findings, given that the study was conducted in selected facilities within two particular sites in each country.

While access to and utilization of antenatal care have improved globally, delivering high-quality ANC remains challenging in LMICs, where the burden of maternal and newborn morbidity and mortality is highest [48,49]. Future research should continue to assess ANC quality, particularly longitudinal quality over the course of pregnancy, with a focus on the appropriateness of risk stratification, disease management, and referrals. In addition, it should examine how longitudinal ANC quality affects adverse perinatal outcomes. Meanwhile, national policies should foster environments that support good clinical practices and the delivery of essential care components that define high-quality ANC.

## Supporting information

**S1 Appendix. Table A.** Description of regression models. Table B. Proportion of pregnant women who received each of the six and all six care components at first ANC visits in four countries by research sites. Table C. Proportion of pregnant women who received each of the six and all six care components at first ANC visits in four countries by facility ownership. Table D. Results of mixed-effect logistic regressions (ANC quality as a continuous score).
(DOCX)

## Author contributions

**Conceptualization:** Wen-Chien Yang, Margaret E. Kruk, Catherine Arsenault.

**Data curation:** Shalom Sabwa, Catherine Arsenault.

**Formal analysis:** Wen-Chien Yang, Shalom Sabwa.

**Funding acquisition:** Margaret E. Kruk.

**Methodology:** Wen-Chien Yang.

**Project administration:** Shalom Sabwa, Anagaw Derseh Mebratie, Beatrice Amboko, Irene Mugenya, Nokuzola Cynthia Mzolo, Nompumelelo Gloria Mfeka-Nkabinde, Theodros Getachew, Tefera Taddele, Damen Haile Mariam, Sailesh Mohan, Prashant Jarhyan, Catherine Arsenault.

**Resources:** Margaret E. Kruk.

**Supervision:** Margaret E. Kruk, Catherine Arsenault.

**Writing – original draft:** Wen-Chien Yang, Catherine Arsenault.

**Writing – review & editing:** Shalom Sabwa, Anagaw Derseh Mebratie, Beatrice Amboko, Irene Mugenya, Sein Kim, Emily R Smith, Monica Chaudhry, Nokuzola Cynthia Mzolo, Nompumelelo Gloria Mfeka-Nkabinde, Theodros Getachew, Tefera Taddele, Damen Haile Mariam, Sailesh Mohan, Prashant Jarhyan, Margaret E. Kruk, Catherine Arsenault.

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
