## [Decision Letter · Decision Letter 0]

11 Feb 2026

PGPH-D-25-02244

Quality of antenatal care and perinatal outcomes: evidence from a cohort study in Ethiopia, Kenya, South Africa, and India

Dear Dr. Arsenault,

Thank you for submitting your manuscript to PLOS Global Public Health. After careful consideration, we feel that it has merit but does not fully meet PLOS Global Public Health’s publication criteria as it currently stands. Therefore, we invite you to submit a revised version of the manuscript that addresses the points raised during the review process.

We look forward to receiving your revised manuscript.

Kind regards,

Nicola Hawley

Academic Editor

Journal Requirements:

1. Please provide a detailed online Financial Disclosure statement. This is published with the article. It must therefore be completed in full sentences and contain the exact wording you wish to be published.

a) State the initials, alongside each funding source, of each author to receive each grant. For example: “This work was supported by the National Institutes of Health (####### to AM; ###### to CJ) and the National Science Foundation (###### to AM).”

For more information, please go to our submission guidelines:

https://journals.plos.org/globalpublichealth/s/submission-guidelines#loc-financial-disclosure-statement

2. Please ensure that the funders and grant numbers match between the Financial Disclosure field and the Funding Information tab in your submission form. Note that the funders must be provided in the same order in both places as well.

3. Please update your online Competing Interests statement. If you have no competing interests to declare, please state: “The authors have declared that no competing interests exist.”

4. In the online submission form, you indicated that your data will be submitted to a repository upon acceptance. We strongly recommend all authors deposit their data before acceptance, as the process can be lengthy and hold up publication timelines. Please note that, though access restrictions are acceptable now, your entire minimal dataset will need to be made freely accessible if your manuscript is accepted for publication. This policy applies to all data except where public deposition would breach compliance with the protocol approved by your research ethics board. If you are unable to adhere to our open data policy, please kindly revise your statement to explain your reasoning and we will seek the editor's input on an exemption.

5. Please provide separate main figure files in .tif or .eps format only and ensure that all files are under our size limit of 10MB.

Additional Editor Comments (if provided):

As you will see from the reviewers written comments, some suggestions are substantive and others are stylistic. As you are preparing your revision, please pay attention to the substantive comments (for example, reviewer #2's comments about making clear that quality is based on the first antenatal care visit) but feel free to use your discretion as you review the more stylistic edits requested.

Reviewers' comments:

Reviewer's Responses to Questions

**Comments to the Author**

1. Does this manuscript meet PLOS Global Public Health’s publication criteria? Is the manuscript technically sound, and do the data support the conclusions? The manuscript must describe methodologically and ethically rigorous research with conclusions that are appropriately drawn based on the data presented.? Is the manuscript technically sound, and do the data support the conclusions? The manuscript must describe methodologically and ethically rigorous research with conclusions that are appropriately drawn based on the data presented.

Reviewer #1: Yes

Reviewer #2: Yes

2. Has the statistical analysis been performed appropriately and rigorously?

Reviewer #1: Yes

Reviewer #2: Yes

3. Have the authors made all data underlying the findings in their manuscript fully available (please refer to the Data Availability Statement at the start of the manuscript PDF file)?

The PLOS Data policy requires authors to make all data underlying the findings described in their manuscript fully available without restriction, with rare exception. The data should be provided as part of the manuscript or its supporting information, or deposited to a public repository. For example, in addition to summary statistics, the data points behind means, medians and variance measures should be available. If there are restrictions on publicly sharing data—e.g. participant privacy or use of data from a third party—those must be specified.requires authors to make all data underlying the findings described in their manuscript fully available without restriction, with rare exception. The data should be provided as part of the manuscript or its supporting information, or deposited to a public repository. For example, in addition to summary statistics, the data points behind means, medians and variance measures should be available. If there are restrictions on publicly sharing data—e.g. participant privacy or use of data from a third party—those must be specified.

Reviewer #1: Yes

Reviewer #2: Yes

4. Is the manuscript presented in an intelligible fashion and written in standard English?

Reviewer #1: Yes

Reviewer #2: Yes

Reviewer #1: Review Reports

Title: Quality of antenatal care and perinatal outcomes: evidence from a cohort study in

Ethiopia, Kenya, South Africa, and India

Manuscript ID: PGPH-D-25-02244

Review Comments

General Comments

We appreciate the team and collaborative effort for the study.

The guideline of the journal should be used well. E.g. Methods and findings in the abstract sub section are not in line with the guideline of the journal.

The format is incorrect

Avoid general words E.g. In line 37-38 In some settings., Again line 39 £ resource limited countries

Grammar issues E.g. Line 39 can be rewritten as This study had assessed

Inconsistency E.g. perinatal outcomes Versus fetal loss and LBW

Some lack full stop and contain incomplete sentence

Specific Comments

Title: Use time frame

Abstract

The gap is weak

Maintain consistency

The result should report confidence interval after the percentage

What about Quality ANC and fetal loss? [Is there association]

The conclusion is not in line with the objective of the study E.g. Quality gaps Versus quality ANC

Introduction

The section fails to contain what it should scientifically contain e.g. What are efforts made globally, regionally and nationally in the respective countries to escalate quality ANC and reduce adverse perinatal outcomes?

What is the importance of stating still birth instead of fetal loss?

Methods

The number of sites per country should be stated.

The detail of the study participants was missed E.g. Age.

When do you say quality ANC? Is that based on the number of visits or have you used quality frameworks to assess it? If so, what type of quality framework and why?

Is that ANC utilization study? Or?

If Ultrasound was used why you take the rough estimate of pregnancy by the pregnant women herself?

The primary outcome lacks reference? E.g. Low birth weight. In addition, you failed also to define the fetal loss and cite appropriate reference.

Line 191-192 We also created a good quality ANC binary variable based on the receipt of five care components (excluding ultrasound scan). Is not in line with the abstract. This is also not in line with the idea in line 248.

How do you manage data from the public health facilities and mixed data from the public and the private health facilities [Comparability and how you treated it?]

Have you used the current recommendation of WHO ANC guideline or Country guideline? Meaning what is the source of the quality ANC at six visits?

What are primary independent variables?

Have you done model fitness?

Results

Present systematically

Sub group analysis E.g. Age

Maintain coherence

Use illustrative E.g For the perinatal outcome

Tables are incomplete E.g Sample size.

Discussion and Other consequent sections

The discussion is weak and needs enrichment

Use your own main findings

Try to write strong theoretical and practical implication with appropriate explanations, justifications and reference.

It lacks recommendations

Regards,

Reviewer #2: The authors are commended for the analysis and the write-up of this manuscript. However, it could be seen that the authors are not really studying "ANC quality" expected to be for the antenatal care received until the end of the study. This research is actually studying "Quality of the first (initial) ANC Assessment" and this should be a key point in the discussion. For the manuscript to be published, I suggest the following as major revision to be addressed by the authors:

1. The Title, Abstract its Conclusion, as well as the Discussion Section must reflect the specific exposure which is the "Quality of the First Antenatal Care Visit." Thus, the word "First" or "Initial" should be used to specify the antenatal quality this research assessed.

2. The Discussion must contain a prominent subsection explicitly detailing the implications of using a first-visit-only quality measure. The authors should state that their study design was not able to test the effect of longitudinal, ongoing care quality, which may be more relevant for outcomes like LBW.

3. It can be recommended that future research should investigate how to maintain high-quality care across all ANC contacts, as this longitudinal quality may be critical for preventing outcomes like low birth weight, which this study was not designed to assess.

4. Reframe the LBW finding: We found no association between the completeness of the first ANC visit and low birth weight. This will not render subsequent antenatal care at other time points irrelevant

**Do you want your identity to be public for this peer review?** For information about this choice, including consent withdrawal, please see our Privacy Policy..

Reviewer #1: No

Reviewer #2: **Yes:** Taiwo AbionaTaiwo AbionaTaiwo AbionaTaiwo Abiona

---

## [Editor Report · Decision Letter 1]

16 Mar 2026

PGPH-D-25-02244R1

Quality of first antenatal care visits and perinatal outcomes: evidence from a cohort study in Ethiopia, Kenya, South Africa, and India

Dear Dr. Arsenault,

Thank you for submitting your manuscript to PLOS Global Public Health. After careful consideration, we feel that it has merit but does not fully meet PLOS Global Public Health’s publication criteria as it currently stands. Therefore, we invite you to submit a revised version of the manuscript that addresses the points raised during the review process.

Thank you for your attention to the comments raised in the prior review. You did an excellent job addressing them.

Please could you also add some text to the background section that explicitly highlights the importance of the first ANC visit? I think that is important now that the paper makes clear that the data collected speaks to that visit specifically.

We look forward to receiving your revised manuscript.

Kind regards,

Nicola L. Hawley

Academic Editor

Journal Requirements:

Additional Editor Comments (if provided):

Thank you for your attention to the comments raised in the prior review. You did an excellent job addressing them.

Please could you also add some text to the background section that explicitly highlights the importance of the first ANC visit? I think that is important now that the paper makes clear that the data collected speaks to that visit specifically.
---

## [Editor Report · Decision Letter 2]

23 Mar 2026

Quality of first antenatal care visits and perinatal outcomes: evidence from a cohort study in Ethiopia, Kenya, South Africa, and India

PGPH-D-25-02244R2

Dear Dr. Arsenault,

We are pleased to inform you that your manuscript 'Quality of first antenatal care visits and perinatal outcomes: evidence from a cohort study in Ethiopia, Kenya, South Africa, and India' has been provisionally accepted for publication in PLOS Global Public Health.

Best regards,

Nicola L. Hawley

Academic Editor